# Cell-accurate optical mapping across the entire developing heart

**Michael Weber[1,2†], Nico Scherf[1,3†], Alexander M Meyer[4], Daniela Panáková[4,5], Peter Kohl[6], Jan Huisken[1,7]\***

[1]Max Planck Institute of Molecular Cell Biology and Genetics, Dresden, Germany; [2]Harvard Medical School, Boston, United States; [3]Max Planck Institute for Human Cognitive and Brain Sciences, Leipzig, Germany; [4]Max-Delbrück-Center for Molecular Medicine in the Helmholtz Association, Berlin, Germany; [5]DZHK (German Centre for Cardiovascular Research), partner site Berlin, Germany; [6]Institute for Experimental Cardiovascular Medicine, University Heart Centre Freiburg - Bad Krozingen, Faculty of Medicine, Albert-Ludwigs University, Freiburg, Germany; [7]Morgridge Institute for Research, Madison, United States

**Abstract** Organogenesis depends on orchestrated interactions between individual cells and morphogenetically relevant cues at the tissue level. This is true for the heart, whose function critically relies on well-ordered communication between neighboring cells, which is established and fine-tuned during embryonic development. For an integrated understanding of the development of structure and function, we need to move from isolated snap-shot observations of either microscopic or macroscopic parameters to simultaneous and, ideally continuous, cell-to-organ scale imaging. We introduce cell-accurate three-dimensional $Ca^{2+}$-mapping of all cells in the entire electro-mechanically uncoupled heart during the looping stage of live embryonic zebrafish, using high-speed light sheet microscopy and tailored image processing and analysis. We show how myocardial region-specific heterogeneity in cell function emerges during early development and how structural patterning goes hand-in-hand with functional maturation of the entire heart. Our method opens the way to systematic, scale-bridging, *in vivo* studies of vertebrate organogenesis by cell-accurate structure-function mapping across entire organs.
DOI: https://doi.org/10.7554/eLife.28307.001

**\*For correspondence:** jhuisken@gmail.com

†These authors contributed equally to this work

**Competing interests:** The authors declare that no competing interests exist.

## Introduction

Organogenesis builds on cell–cell interactions that shape tissue properties, and tissue-level cues that control maturation of cell structure and function. During cardiogenesis, region-specific heterogeneity in cellular activity patterns evolves as the heart undergoes large-scale morphological changes: The spontaneously active heart tube develops into the mature heart, in which pacemaker cells near the inflow site initiate the rhythmic excitation that spreads with differential velocities through distinct regions of the myocardium. This controlled cardiac activation gives rise to an orderly sequence of atrial and ventricular calcium release and contraction. An integrated understanding of cardiogenesis at the systems level requires simultaneous cell and organ scale imaging, under physiological conditions *in vivo*. Here, we present a high-speed light sheet microscopy and data analysis pipeline to measure fluorescent reporters of cardiomyocyte location and activity across the entire electro-mechanically uncoupled heart in living zebrafish embryos during the crucial looping period from 36 to 52 hr post fertilization (hpf). By noninvasively reconstructing the maturation process of the myocardium in its entirety at cellular resolution, our approach offers an integrative perspective on tissue and cell levels simultaneously, which has previously required separate experimental setups and

**eLife digest** The heart has a built-in pacemaker that sets the rhythm of the heartbeat. Pacemaker cells produce electrical signals that spread across the heart in a coordinated wave. As each cell receives its signal, ion channels open in its membrane. Calcium ions rush in from the spaces around the cells, triggering the release of more calcium ions from internal stores. The rise in calcium ion levels causes the heart muscle to contract.

Standard techniques for studying how the activation process spreads across the heart typically involve removing the organ from the animal. One reason for this is that no microscopy technique had been able to provide the detail needed to observe the activity of individual cells across the whole heart during its activation cycle.

Zebrafish embryos have a simple heart with two chambers that can be visually explored because the embryos are transparent. Their hearts are activated in a pattern that has been maintained throughout evolution with principal similarities in many different species. These properties make fish embryos well suited for the non-invasive examination of the heart.

Weber, Scherf et al. have studied genetically engineered zebrafish embryos whose heart muscle cells contained a calcium-sensitive fluorophore, using a technique called light sheet microscopy. This method illuminates the heart with a thin sheet of laser light, which causes the fluorescent dye to glow in a way that indicates changes in the concentration of calcium ions in the cells. A fast and sensitive camera detects these signals and stacks of movies are recorded and synchronized, allowing cardiac activation to be mapped in three dimensions as it spreads across the heart.

Applying this new technique revealed that different parts of the heart conduct activation signals at different speeds. These speeds finely match the anatomical features of the heart, yielding planar progression of the activation signal over the increasingly complex shape of the developing heart. Weber, Scherf et al. also showed that the heart only requires a handful of pacemaker cells to reliably set the heart's rhythm.

Future modifications to the technique of Weber, Scherf et al. could help us investigate how the heart works in even finer detail. For example, it might reveal how electrical activity, calcium handling, and contraction influence one another, and how they individually and collectively respond to drug treatments. This will help us understand how the normal heart rhythm develops, how it can be modified, and how the heart adapts to changes in its environment, including damage during cardiac disease.

DOI: https://doi.org/10.7554/eLife.28307.002

specimens. Our method opens a new way towards systematic assessment of the mutual interrelations between cell- and tissue properties during organogenesis.

## Results

The zebrafish is an appealing vertebrate model system with a simple, yet functionally conserved heart. Light sheet microscopy has proven to be supremely suited for obtaining *in vivo* recordings of the intact embryonic zebrafish heart (*Chi et al., 2008*; *Scherz et al., 2008*; *Arnaout et al., 2007*; *Trivedi et al., 2015*). Whole cardiac cycles have been reconstructed in 4D (3D + time) using post-acquisition synchronization of high-speed light sheet movies in a z-stack. The resulting effective temporal resolution of about 400 volumes per second (*Mickoleit et al., 2014*) is unmatched by other *in vivo* volumetric imaging techniques such as light sheet microscopy with electrically focus-tunable lenses or swept, confocally-aligned planar excitation (*Bouchard et al., 2015*; *Fahrbach et al., 2013*; *Hou et al., 2014*; *Liebling et al., 2005*). We built a light sheet microscope tailored for high-speed imaging of the heart in the living zebrafish embryo. By fine-tuning the magnification and restricting camera readout to the center area of the chip, we balanced the field of view and the spatial and temporal sampling to record cardiac activation in the entire heart with cellular precision (Materials and methods).

We investigated whether post-acquisition synchronization could be extended to visualizing calcium transients in cardiac myocytes across the entire heart of living embryonic zebrafish expressing the fluorescent calcium reporter GCaMP5G under the *myl7* promoter (*Figure 1a*, *Figure 1—figure*

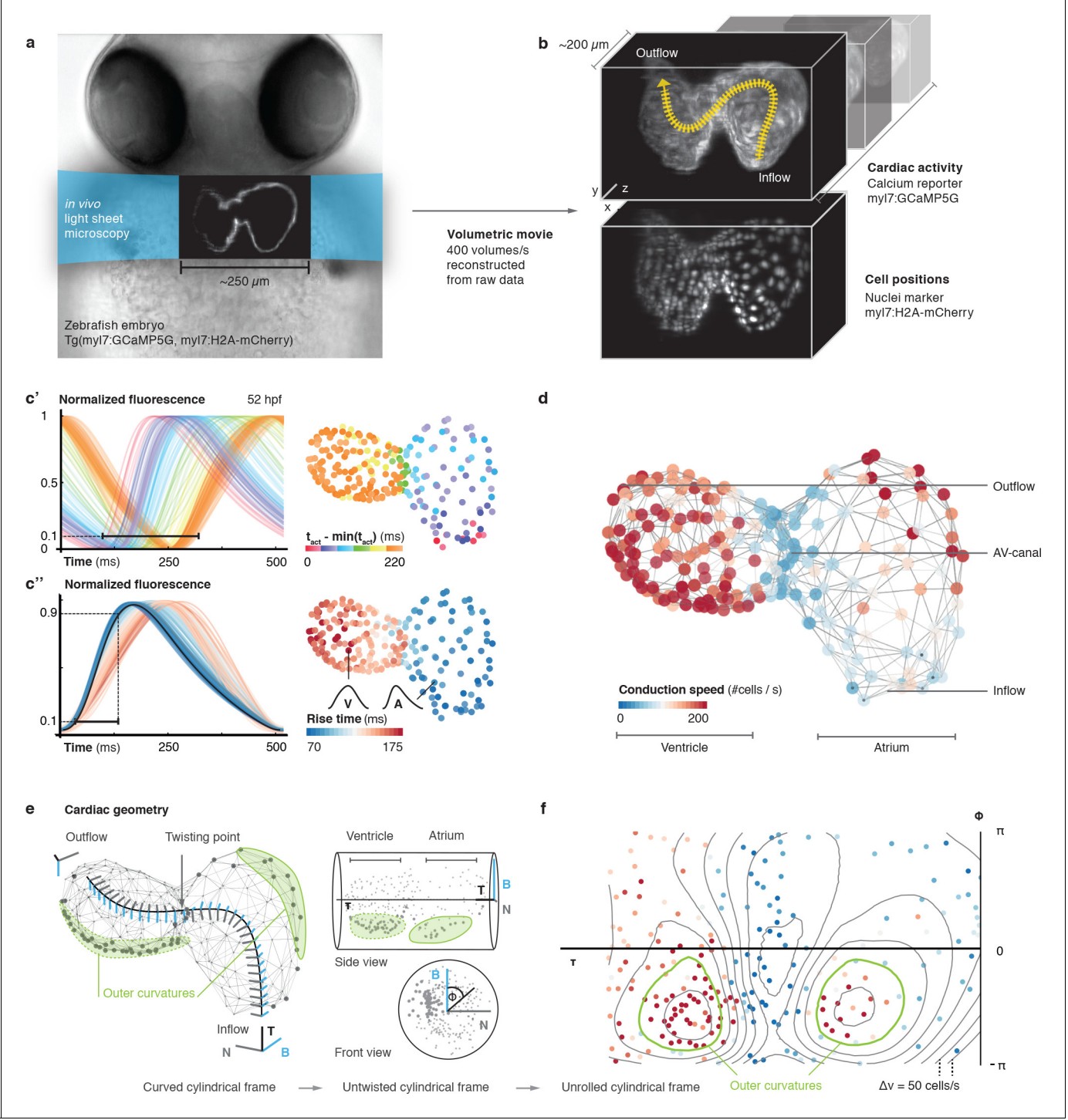

**Figure 1.** *In vivo* 3D optical mapping reveals cell-specific calcium transient patterns at 52 hr post fertilization (hpf). (**a**) Transmitted light microscopy image with ~250 µm-sized, two-chambered heart (shown as fluorescence image with light sheet illumination path). (**b**) Genetically encoded fluorescent markers expressed in myocardial cells report calcium transient activity and cell positions. Volumetric movies were reconstructed from multiple high-speed movies, each with a temporal resolution of 2.5 ms and a voxel size of 0.5 µm in *xy* and 1 µm in *z*. Image data are available at *Weber et al. (2017)*. (**c'**) Normalized fluorescence plot of every cell's calcium transient over one cardiac cycle. The network's activation timing ($t_{act}$) is visualized in 3D based on the time-point of 10% calcium transient amplitude in every individual cell (right, same color scale). (**c''**) Normalized fluorescence plot of all calcium transients, aligned in time based on the timing of deviation from minimal fluorescence intensity (3D network, same color scale). (**d**) Biological conduction speed, expressed as cells activated per unit of time, is visualized on the 3D network. (**e**) The basis vectors of the local coordinate system (tangent – black, normal – grey, and binormal – blue) are shown, moving along the centerline. The initial and final orientation of the moving reference

*Figure 1 continued on next page*

*Figure 1 continued*

frame is shown in a zoomed version at inflow and outflow sites. Outer curvature regions of atrium and ventricle are highlighted in green. (f) The unrolled cylinder results in a representation of 3D network function in a 2D map. Projections show conduction speed across the entire myocardium; iso-velocity lines ($\Delta v$ = 50 cells/s) are shown in black. Outer curvature regions in atrium and ventricle are highlighted in green.

DOI: https://doi.org/10.7554/eLife.28307.003

The following figure supplements are available for figure 1:

**Figure supplement 1.** High-speed light sheet microscopy for *in vivo* 3D optical mapping.

DOI: https://doi.org/10.7554/eLife.28307.004

**Figure supplement 2.** Comparison of the calcium reporter GCaMP5G and the voltage reporter Arch(D95N) for multi-scale readout of cardiomyocyte activation.

DOI: https://doi.org/10.7554/eLife.28307.005

**Figure supplement 3.** Evaluation of signal coverage and tracing precision using the fluorescent calcium reporter GCaMP5G at different recording speeds.

DOI: https://doi.org/10.7554/eLife.28307.006

**Figure supplement 4.** Activation and conduction properties of cells vary from inflow to outflow, with patterns conserved across different hearts (52 hpf, n = 3).

DOI: https://doi.org/10.7554/eLife.28307.007

**Figure supplement 5.** Reconstruction of myocardial topology.

DOI: https://doi.org/10.7554/eLife.28307.008

**Figure supplement 6.** Comparison of metric and cell-to-cell speed measurements.

DOI: https://doi.org/10.7554/eLife.28307.009

**Figure supplement 7.** Location of pacemaker cells.

DOI: https://doi.org/10.7554/eLife.28307.010

**Figure supplement 8.** Myocardial morphology is heterogeneous across the heart at 52 hpf.

DOI: https://doi.org/10.7554/eLife.28307.011

**Figure supplement 9.** Patterns of cell activation across myocardium.

DOI: https://doi.org/10.7554/eLife.28307.012

*supplement 1*). The genetically expressed calcium reporter provides a specific, consistent and non-invasive readout of cardiomyocyte activity *in vivo* (*Figure 1b*, *Videos 1* and *2*). In a side-by-side comparison, the calcium signal had good and stable fluorescent yield at low excitation power, superior to genetically expressed voltage reporters. Importantly, the calcium signal faithfully reports presence and timing of cell activation (*Figure 1—figure supplement 2*) (*Kralj et al., 2011*). To prevent interference of tissue movement and deformation with observed signals, we decoupled electrical excitation and mechanical contraction by inhibiting the formation of the calcium-sensitive regulatory complex within sarcomeres, using a morpholino against *tnnt2a* (Materials and methods). By mounting zebrafish embryos in low concentration agarose inside polymer tubes, we could position the embryos for precise optical investigation without anesthesia (*Figure 1—figure supplement 1a,b*). To attribute calcium dynamics to individual cardiomyocytes, we also recorded a fluorescent nuclear marker (*myl7*:H2A-mCherry). The high temporal (400 Hz) and spatial sampling (0.5 μm pixel size) was adequate for computing normalized average calcium transients throughout the cardiac cycle for each cell across the entire heart (*Figure 1—figure supplement 3*, *Video 3*, Materials and methods).

To map cellular activation timings onto a 3D structural representation of the heart, we

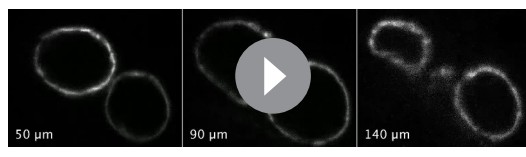

**Video 1.** Raw GCaMP5G signal at different imaging depths. Fluorescence signal recorded at 400 Hz at three different depths (50, 90 and 140 μm along the optical axis) in the heart of a living Tg(*myl7*:GCaMP5G) zebrafish at 52 hpf using the described imaging setup.

DOI: https://doi.org/10.7554/eLife.28307.013

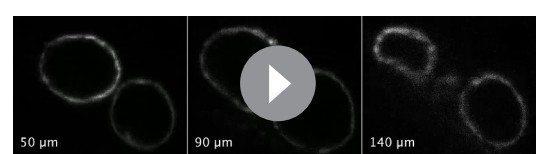

**Video 2.** Calcium signal at different imaging depths after synchronization. Raw data from *Video 1* after completed post-acquisition synchronization.

DOI: https://doi.org/10.7554/eLife.28307.014

identified every single cell's activity (*Figure 1c'*) and quantified dynamic characteristics of all cardiac myocytes across space and time. The distribution of calcium transient rise times (from 10% to 90% of calcium transient peak amplitude) revealed the emergence of distinct upstroke characteristics in different locations within the 3D network (by 52 hpf). Rise times were shortest for atrial muscle cells, intermediate in the atrio-ventricular canal (AVC), and longest for ventricular cardiomyoctes (*Figure 1c''*), in keeping with higher vertebrates where atrial myocyte contraction is faster than that of ventricular cells (*Brandenburg et al., 2016*).

To get a more quantitative understanding of the 3D distribution of activity patterns, a canonical description of cell locations was needed. We traced and parameterized the myocardium's centerline (*Figure 1—figure supplement 4a*, Materials and methods), along which we assigned a unique position to each cell between inflow and outflow. We identified a positive correlation between rise time and position of cells along the midline, with a clear discontinuity at the AVC (*Figure 1—figure supplement 4b*), illustrating emergence of chamber-specific patterns of individual cell activation properties across the heart.

Next, we studied the spatial patterns of sequential cell activation, as a read-out for the speed of electrical conduction across the heart. While cardiac activation in the early linear heart tube is slow and near-uniform, the chambered heart shows areas of elevated conduction velocity (*Dehaan, 1961*; *de Jong et al., 1992*; *Moorman et al., 1998*; *Chi et al., 2008*; *Panáková et al., 2010*). Common imaging-based methods for determining cell conduction speed tend to deliver only a metric, or 'biophysical speed' of conduction (distance over time). To reflect biological progression of activation between cells (of potentially different or – in contracting tissue – dynamically changing size), we assessed local cell topology across the entire heart (*Figure 1d*, *Figure 1—figure supplement 5*, Materials and methods) to also calculate the 'biological speed' (number of activated cells over time) of conduction (*Video 4*). Our analysis revealed that biophysical and biological conduction velocities show differences between and within anatomical regions of the heart (*Figure 1—figure supplement 6*). Either descriptor identifies particularly slow conduction between the most proximal atrial cells and between the cells of the AVC, while faster conduction is seen among working myocardial cells of the atrium and ventricle. In the atrium, there is a bias towards higher biophysical speeds, due to larger cell dimensions (*Figure 1d*, *Figure 1—figure supplement 4c*).

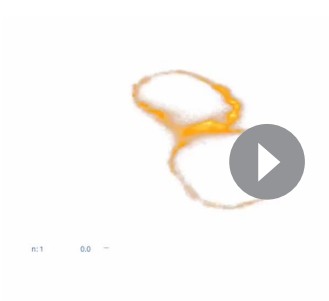

**Video 3.** 4D reconstruction of cellular calcium transients. (1) 4D reconstruction of calcium activation from the synchronized planar movies recorded in the heart of a living Tg(*myl7*:GCaMP5G) zebrafish embryo at 52 hpf. The number of slices used in the reconstruction is indicated by n. Raw GCaMP5G signal is shown in orange. Time is indicated in ms. (2) Cell detection: The raw fluorescence signal of *myl7*:H2A-mCherry is overlaid with the centroids of detected nuclei. (3) Calcium mapping: The raw *myl7*:GCaMP5G signal is shown. The dots indicate the extracted cellular positions. (4) Cell-specific transients: The normalized GCaMP5G transients are shown for two sample cells (one atrial and one ventricular cell).

DOI: https://doi.org/10.7554/eLife.28307.015

The activation sequence in the pacemaker region is of particular importance to heart physiology and function, yet individual cells that give rise to earliest activation were difficult to identify with previous methods (*Van Mierop, 1967*; *Arrenberg et al., 2010*; *Christoffels et al., 2010*; *Tessadori et al., 2012*). We found that, at 52 hpf, less than 10 cells per heart serve as activation origins. They are located in the sinus venosus at the heart's inflow side, which is a homologue of the primary cardiac pacemaker region in adult heart (*Poon and Brand, 2013*). Our data further show that the ring-like arrangement of pacemaker cells (*Figure 1—figure supplement 7*), together with a preferential orientation of myocardial cells in that region perpendicular to the inflow-outflow direction (*Auman et al., 2007*) (*Figure 1—figure supplement 8*), generates the initial planar 'ring-like' activation front that propagates evenly into the atrium.

To visualize the 3D conduction pattern, the heart can be represented as a curved cylinder with varying diameter and s-shaped deformations in the coronal and in the sagittal plane. We used the position along the midline ($\tau$) and the local Frenet-Serret frame as intrinsic coordinates ($\Phi$, z)

(*Figure 1e* and *Video 5*, Materials and methods). Interestingly, by following the orientation of this reference system along the midline, we noticed torsion associated with the cardiac looping, which is most pronounced around the AVC (*Männer, 2000*; *Christoffels et al., 2000*; *Harvey, 2002*). Establishing the intrinsic coordinates of the curved cylinder allows straightening and untwisting by implicitly removing the morphological torsion. Neglecting the actual distance to the midline and plotting the position along the midline (τ) against the angle (Φ), we obtained a 2D projection (*Figure 1e,f* and *Video 5*, Materials and methods), in which the two outer curvature regions of atrium and ventricle are located side-by-side. A clear asymmetry in conduction speeds between cells at the inner and the outer curvatures was apparent in this representation (*Figure 1f*). Irrespective of these differences, however, the activation wave travelled smoothly in a ring-like fashion along the heart, as indicated by the isochronal lines in the cylindrical projection (*Figure 1—figure supplement 9*).

In order to document how heterogeneity in cardiac function arises during cardiac looping, we extended our 3D optical mapping towards earlier developmental stages. During the crucial period between 36 and 52 hpf, ventricular cell number increased by about 45%, the initial heart tube developed into a two-chambered organ, the midline of the heart became increasingly curved and twisted (*Figure 2—figure supplement 1*), and the activation frequency increased – all signs of organ maturation. In spite of a net increase in cell numbers, the time required for activation to propagate from inflow to outflow decreased. A progressive crowding of calcium transient activation dynamics indicated maturation of cells, with two groups differentiating from the early more homogeneous pattern: working cardiomyocytes in atrium and ventricle (*Figure 2a–f*). At 36 hpf, activation propagated evenly across the myocardium, compatible with peristalsis. Subsequently, calcium dynamics became increasingly structured. By 52 hpf, propagation of activation was fast across atrium and ventricle, while it remained slow in the AVC (*Figure 2—figure supplement 2a*). With organ maturation during these 16 hr, cells in the ventricular part of the network showed longer calcium transient rise time

(*Figure 2—figure supplement 2b*) but faster inter-cellular spread of activation (*Figure 2g–l*, *Figure 2—figure supplement 2c*). Conduction also changed within chambers, such as along the outer curvature in the atrium. Increasing deformation and twisting of the cardiac tissue was associated with changes in cell shape (cf. *Figure 2d–f*)

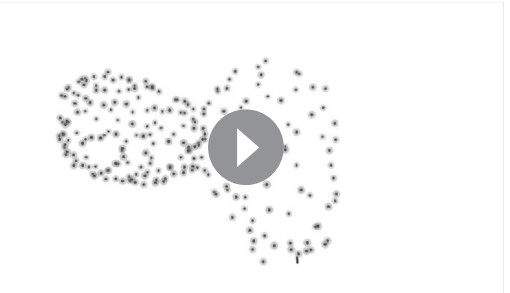

**Video 5.** Mapping of myocardial geometry. (1) Tracing of midline: The black line traces the center of the myocardium from inflow to outflow. Cellular positions are shown as gray spheres. (2) Moving reference frame: The intrinsic reference frame is shown along the extracted midline. The three axes represent the tangent (black), normal (gray), and binormal (blue) vectors. The trace of the binormal vector is shown as blue band behind the moving trihedron. (3) Untwisting: The heart is successively untwisted by reducing the torsion of the midline to 0. The gray lines indicate traces of cells during this process. (4) Straightening: The heart is straightened by reducing the remaining curvature of the midline to 0. (5) Projection to cylinder. Each cell is projected to the same radial distance from the straightened midline resulting in a cylindrical reference system. (6) Unrolling: The cylinder is unfolded into the 2D plane, resulting in the 2D plots used in the main text. Color code in the final frame indicates biological conduction speed.
DOI: https://doi.org/10.7554/eLife.28307.017

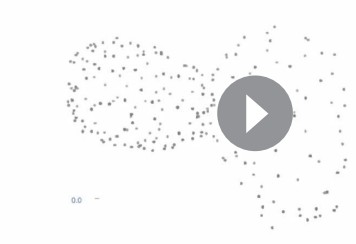

**Video 4.** Computation of biological conduction speed. (1) Activation timing: Cell positions are indicated by gray dots. Activated cells are highlighted in yellow. Time is shown in ms. (2) Network reconstruction: Estimated local topology is shown as gray edges connecting neighboring cells. (3) Activation across network: Activated cells in the network are highlighted in yellow. (4) Cell–cell conduction speed: The average conduction speed is shown for each cell after activation of its neighbors in the network. Color code indicates conduction speed (red - high, blue - low).
DOI: https://doi.org/10.7554/eLife.28307.016

 

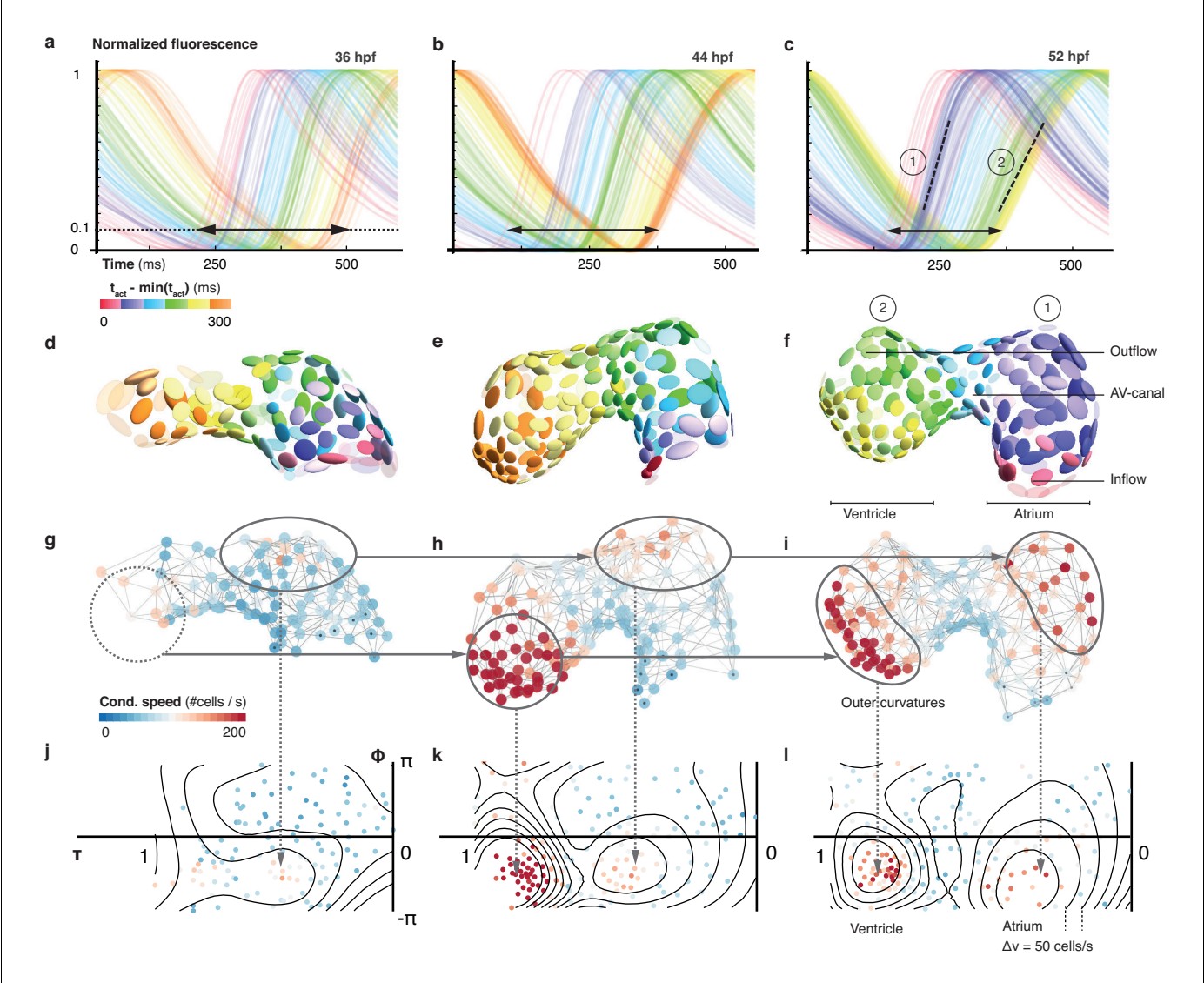

**Figure 2.** Organ maturation and functional cell remodelling from 36 to 52 hpf. (a–c) Normalized calcium transients of all cells are shown at three different time-points: (a) 36, (b) 44, and (c) 52 hpf. The shortening of black arrows indicates the decrease in time between earliest and latest activation of cells along the whole heart. The gradual formation of atrial (1) and ventricular (2) populations with increasingly different calcium transients is highlighted by the dashed lines at 52 hpf. The color of each transient indicates its activation time (all three plots use the same color scale). (d–f) Changes in cardiac and cellular morphology at each developmental time point. Estimated cell shapes are visualized as scaled ellipsoids. The color of each ellipsoid indicates activation time (relative to the first-activated cell); color code as in (a–c). Two main populations of cells corresponding to highlighted clusters in (c) are indicated by numbers. (g–i) 3D pattern of biological conduction speeds across the heart. The formation of outer curvature clusters of cells over time is highlighted by arrows. (j–l) Development of conduction speeds across the myocardium, shown as 2D projections for the three time points. The color code indicates conduction speed. Isovelocity lines are shown in black ($\Delta v$ = 50 cells/s). Corresponding outer curve cell populations are highlighted by arrows.

DOI: https://doi.org/10.7554/eLife.28307.018

The following figure supplements are available for figure 2:

**Figure supplement 1.** Structural organ maturation from 36 to 52 hpf.

DOI: https://doi.org/10.7554/eLife.28307.019

**Figure supplement 2.** Developmental changes of cellular characteristics along the heart.

DOI: https://doi.org/10.7554/eLife.28307.020

**Figure supplement 3.** Cell shape changes during development.

DOI: https://doi.org/10.7554/eLife.28307.021

and in conduction (*Figure 2—figure supplement 3*).

## Discussion

By noninvasively reconstructing the maturation process of the myocardium in its entirety at cellular resolution, our approach offers an integrative perspective on tissue and cell levels simultaneously. We show functional maturation in line with structural patterning of the heart muscle during development: starting from similar initial states, functionally distinct characteristics of calcium transients and conduction properties develop with a highly reproducible pattern relative to the cell locations (cf. *Figure 1—figure supplement 8*). Myocardial cells in the two chambers remodel and specialize into functional tissue of working atrial and ventricular cardiomyocytes, while cells in the pacemaker and AVC regions continue to resemble the earlier phenotype from the tubular stage.

We demonstrate that myocardial activity can be recorded and analyzed with cellular detail across the entire embryonic heart. Future technological advancements can extend the scope of our approach: First, genetically expressed voltage reporters with improved dynamic range may provide a direct readout of myocardial electrical activity. Second, cameras with higher speed and sensitivity would enhance the recording frequency of rapid volume scanning, needed to explore aberrant myocardial activation during arrhythmias. Third, the integration of an algorithm capable of tracking cells in 3D during cardiac contractions would allow investigations in fully functional hearts. Fourth, the addition of optically gated actuators, such as light-activated ion channels or photo-pharmacological probes, would enable contact-free stimulation to probe the roles of individual cells or groups of cells in pacemaking, conduction, and arrhythmogenesis.

Our work further highlights the value of the zebrafish as a vertebrate model system for *in vivo* cardiology, especially when combined with high-speed light sheet microscopy and suitable data analysis pipelines. It opens the way to systematic, scale-bridging, *in vivo* studies of organogenesis by facilitating cell-accurate measurements across entire organs.

## Materials and methods

### Fish husbandry and lines

Zebrafish (*Danio rerio*) were kept at 28.5°C and handled according to established protocols (*Nusslein-Volhard and Dahm, 2002*) and in accordance with EU directive 2011/63/EU as well as the German Animal Welfare Act. Transgenic zebrafish lines Tg(*myl7*:GCaMP5G-Arch(D95N)) (*Hou et al., 2014*), Tg(*myl7*:H2A-mCherry) (*Schumacher et al., 2013*) and Tg(*myl7*:lck-EGFP)$^{md71}$ were used. The lck sequence was PCR amplified from pN1-Lck-GCaMP3 (Addgene, #26974) with In-Fusion primers 5'-GCAAAAGATCTGCCACCATGGGCTGTGGCTGC-3' (forward) and 5'-GCAAAGGGCCCCGAGA TCCTTATCGTCATCGT-3' (reverse) designed with http://bioinfo.clontech.com/infusion/convertPcr-PrimersInit.do and cloned into pEGFP-N1 (Clontech, #6085–1). PCR product generated from attB-flanked BP primers 5'-GGGGACAAGTTTGTACAAAAAAGCAGGCTGGATGGGCTGTGGCTGCAGC TCAAACC-3' (forward) and 5'-GGGGACCACTTTGTACAAGAAAGCTGGGTCTTACTTGTACAGC TCGTCCATGCCGAG-3' (reverse) was BP Clonase II cloned into Gateway pDONR221 (ThermoFisher Scientific, #12536017) to generate the middle entry clone that was further assembled with p5E_myl7, p3E_SV40polyA (Tol2kit #302), and pDEST.Cryst.YFP$^{76}$ (*Mosimann et al., 2015*) into Tol2 transgene plasmid using MultiSite Gateway assembly. See supplementary files for detailed digital plasmid maps of these vectors. To generate Tg(*myl7*:lck-EGFP)$^{md71}$ Tol2-mediated zebrafish transgenesis was performed by injecting 25 ng/ml transgene plasmid together with 25 ng/ml capped *Tol2* transposase mRNA, followed by subsequent screening of positive F0 founders. During the 1 cell stage, embryos were injected with morpholinos against *tnnt2a* to uncouple electrical and mechanical activity (*Sehnert et al., 2002*).

### Sample preparation for light sheet microscopy

Before imaging, fluorescent embryos were selected for absence of cardiac malformations and contractions, using an Olympus stereomicroscope equipped with an LED for transmitted light microscopy and a metal-halide light source and filter sets that match the excitation and emission spectra of GCaMP5G and mCherry for fluorescence excitation. Embryos were mounted in either 0.1 or 1.5%

low gelling temperature agarose (Sigma A9414) inside cleaned polymer tubes (FEP tubing, inner/outer diameter 0.8/1.6 mm, BOLA S1815-04).

## Light sheet microscopy

We built a light sheet microscope for *in vivo* cardiac imaging in zebrafish embryos, based on a previously published design (*Mickoleit et al., 2014*). Imaging was performed in live zebrafish embryos between 36 and 52 hr post-fertilization (hpf) at a temperature of 24°C. Heart rate at this temperature is 2 Hz, about 0.5 Hz lower than at the temperature recommended for breeding of 28.5°C (*Baker et al., 1997*; *Kopp et al., 2005*). Embryos were kept in a custom imaging chamber filled with E3 fish medium and illuminated with a static light sheet generated from Coherent Sapphire LP lasers (488 and 561 nm) using a cylindrical lens and a Zeiss 10x/0.2 air illumination objective. Laser power was kept at or below 2 mW in the field of view (measured at the back aperture of the illumination objective) to exclude thermal effects on heart rate (an increase in heart rate was detected at laser powers of 5 mW and above). Fluorescence was collected and recorded using a Zeiss W Plan-Apochromat 20x/1.0 objective, a Zeiss 0.63x camera adapter, a Hamamatsu W-View image splitter and a Hamamatsu Flash 4.0 v2 sCMOS camera. Embryos were held in place by a Zeiss Lightsheet Z.1 sample holder and oriented using motorized translation and rotation stages (Physik Instrumente GmbH, Karlsruhe, Germany). For imaging of GCaMP5G, a z-stack of movies covering the entire heart was recorded by moving each embryo through the light sheet (488 nm excitation, band-pass 525/50 nm emission filter, 2.5 ms exposure time = 400 Hz, 600 frames = 1.5 s/movie, 1 μm z-steps). To ensure an efficient recording and the best possible synchronization (see below), the embryo was rotated by about 30 degrees, such that both atrium and ventricle were visible in the imaging plane for the majority of the z-stack. For imaging of H2A-mCherry, a matching z-stack was recorded immediately afterwards (561 nm excitation, long-pass 565 nm emission filter, 20 ms exposure time = 50 Hz, 1 μm z-steps). Image acquisition was controlled by a custom program written in LabView (National Instruments). Images were streamed onto a RAID-0 array of four 512 GB solid-state drives.

## Data analysis

### Synchronization of z-stacks of movies

The recorded z-stacks of movies were synchronized in time starting at the middle plane and iterating in two independent threads towards first and last plane, respectively. A full cardiac cycle was selected randomly in the middle plane, and Pearson's correlation coefficients were calculated for every cycle in the adjacent plane. The cycle with the highest correlation was selected as the best fit. As demonstrated before, the presence of both atrial and ventricular myocardium in the majority of planes minimized the risk of false synchronization (*Mickoleit et al., 2014*).

### Visualization of synchronized movies

The synchronized movies were visualized (*Video 2*) using custom scripts (*Scherf, 2017*) for Mathematica 11.1 (Wolfram Research Inc., Champaign, Illinois). The raw calcium signal stacks for each time point were normalized and visualized in 3D using direct volume rendering of the resulting intensities. The 3D volume visualizations were then projected to 2D using a standard perspective transform from a defined viewpoint. Animations were created by a smooth interpolation between different viewpoints.

### Segmentation of cell nuclei

Volumes of cell nuclei were extracted from the H2A channel of the raw image volumes using gray-scale blob detection (*Lindeberg, 1998*). To resolve potential errors in automated processing (false detection, missing cells), cell positions were manually curated using an in-house software (*Scherf, 2017*) facilitating evaluation, addition and deletion of nuclear positions in the 3D data sets. Curated nuclei positions were used for further processing.

### Signal extraction and processing

From the measured nuclear positions, a reference volume was extracted for each cell by computing the Euclidean distance transform and thresholding the distance field around each centroid (nucleus) at a radius of 5 voxels. For each thus identified 'cell volume', a calcium transient was extracted by

averaging the signal over all voxel within the reference volume at each time step. The resulting transients were processed using a low pass filter (with a cutoff frequency of 5 Hz) to reduce noise, and finally normalized to the range [0,1].

## Extraction of midline

The heart's midline was manually traced using an in-house software. The user could draw the midline in two different 2D projections (front and top view) by placing discrete points that were connected across the different projections. The final midline was then obtained by a 4$^{th}$ order B-Spline interpolation to facilitate subsequent computation of intrinsic geometric characteristics such as curvature and torsion.

## Frenet-Serret frame

To describe each cell's position within the myocardium, we established a curved cylindrical coordinate system. The position along the midline ($\tau$) was calculated for each cell by finding the closest point on the midline for the respective centroid. The moving coordinate system along the midline was then computed using the Frenet-Serret frame (**Kreyszig, 1959**), calculating the local tangent ($\vec{T}$), normal ($\vec{N}$), and binormal ($\vec{B}$) vectors:

$$\vec{T}(t) = \frac{\vec{r}(\tau)}{|\vec{r}(\tau)|}$$

$$\vec{N}(t) = \frac{\vec{T'}(\tau)}{|\vec{T'}(\tau)|}$$

$$\vec{B}(t) = \vec{T}(\tau) \times \vec{N}(\tau)$$

with $\vec{r}(\tau) \in R^3$ being the actual 3D position vector on the midline parameterized by $\tau$. Then each cell's position was mapped to this new coordinate system as ($\tau$, $\Phi$, z), where $\tau$ was the position along the midline (normalized to 0 and 1 after reparameterization), $\Phi$ the angle with respect to the local binormal vector (in the local **N-B** plane), and z the radial distance of the centroid from the midline.

## Cylindrical 2D map projections

To map cell positions from 3D to 2D, we projected the curved cylindrical coordinates to the first two components ($\tau$, $\Phi$, $z$) $\rightarrow$ ($\tau$, $\Phi$), discarding the radial distance from the midline. This projection facilitates global visualization of the myocardium, irrespective of its looping stage, in a single flat projection by cutting the cylinder at $\Phi = \pm\pi$ and plotting the coordinates in a rectangular coordinate system in the range $\tau \in [0, 1]$ and $\Phi \in [-\pi, \pi]$, respectively.

## Topology reconstruction

The myocardial topology (local connectivity of cells) was estimated by projecting the 3D centroid positions of each cell and its nearest neighbors locally into the 2D Euclidean plane using Singular Value Decomposition (**Bronshtein, 2007**) of the 3D centroid positions. In each local 2D projection, the cellular connectivity was estimated by computing the 2D Delaunay triangulation (**Delaunay, 1934**). The estimated edges $e_{ij}$ between cells and the original 3D centroid positions of each cell $v_i$ were used to abstractly represent the myocardium as an undirected graph $G = (V, E)$ where $v_i \in V, \ e_{ij} \in E$.

## Computation of biological conduction speed

As the spread of activation in cardiac tissue is governed by cell activation and time delay at gap junctions between cells, we used the graph representation to estimate the average local cell-to-cell conduction speed $cs_i$ in terms of (dimensionless) links traversed per unit of time. Thus, at each cell's position the (harmonic) mean of the number of traversed edges per time was computed over all paths from the cell to its neighbors:

$$cs_i = n \left( \sum_j \frac{dt(i,j)}{e(i,j)} \right)^{-1}$$ for all neighbors $j$ of cell $i$ with $dt(i,j)$ being the temporal difference in activation time and $e(i,j)$ the number of edges between cell $i$ and cell $j$.

## Computation of metric conduction speed

For comparison, the metric conduction speed at each cell's position, the (harmonic) mean of the distance travelled by the activation per time, were computed over all paths from the cell to its neighbors:

$$cs_i = n \left( \sum_j \frac{dt(i,j)}{d(i,j)} \right)^{-1}$$ for all neighbors $j$ of cell $i$ with $dt(i,j)$ being the temporal difference in activation time and $d(i,j)$ the 3D Euclidean distance (along the 'surface' defined by the neighborhood graph) between cell $i$ and cell $j$.

## Estimation of cell size and shape

We further used the graph representation to extract an estimate of a cell's shape by computing the principal components of the distribution of neighboring nuclei around each cell. Thus, each cell is represented as an ellipsoidal region defined by the eigenvectors $v_i$ and eigenvalues $\lambda_i$ of the local principal value decomposition. Cell shape was then approximated as the fractional anisotropy of each ellipsoidal region:

$$\sqrt{\frac{3}{2}} \frac{\sqrt{\left(\lambda_1 - \lambda\right)^2 + \left(\lambda_2 - \lambda\right)^2 + \left(\lambda_3 - \lambda\right)^2}}{\sqrt{\lambda_1^2 + \lambda_2^2 + \lambda_3^2}}$$ , with $\lambda$ being the average eigenvalue.

Cell size was approximated as the volume $\frac{4}{3}\pi\lambda_1\lambda_2\lambda_3$ of each ellipsoid.

## Pacemaker identification

Pacemaker cells were defined as the cells that showed earliest activation (activation time shorter than the 0.05-quantile of overall distribution of activation times) and that conducted slowly (conduction speed slower than the median conduction speed within the cell population). All cells falling within this category were labeled as potential pacemaker cells.

## Sample preparation for confocal and two-photon microscopy

Transgenic zebrafish embryos expressing *myl7*:H2A-mCherry and *myl7*:lck-eGFP in the myocardium were screened for absence of cardiac malformations and contractions using an Olympus stereomicroscope at 2 dpf. Selected embryos were mounted in 1.5% low-melting point agarose (Sigma A9414) inside glass capillaries using plungers (Brand 20 µl Transferpettor caps and piston rod). After a few minutes, mounted embryos were carefully transferred onto a custom 3D-printed sample holder with their heart facing up. Once positioned, they were fixed in place using drops of agarose at both ends of the agarose column. The sample holder was placed in a 52 mm plastic dish and covered with E3 fish medium.

## Confocal and two-photon microscopy

Confocal and two-photon microscopy was performed with an upright Zeiss LSM 780 NLO equipped with a Zeiss W Plan-Apochromat 20x/1.0 objective lens, a Coherent Chameleon multiphoton laser set to 920 nm, a Coherent HeNe 594 nm laser and gallium arsenide phosphide (GaAsP) detectors. The fluorescence reporter *myl7*:H2A-mCherry was recorded using single-photon excitation at 594 nm, a descanned GaAsP detector and a confocal pinhole set to one airy unit. Two-photon excitation at 920 nm and a non-descanned GaAsP detector were used for the fluorescence reporter *myl7*:lck-eGFP. The pixel size was 0.148 µm$^2$ and z-stacks were recorded with a 1 µm step size. The total acquisition time of one z-stack was about 60 min.

## Acknowledgements

We thank the entire Huisken Lab, in particular R Power and M Mickoleit, as well as A El-Armouche, VM Christoffels, DJ Christini, F Ortega, L Herzel and K Thierbach for valuable discussions. Special thanks to S Bundschuh for support with confocal and two-photon microscopy. NS and JH are

supported by the ERC Consolidator Grant SmartMic and PK is supported by the ERC Advanced Grant CardioNECT. AMM and DP are supported by the Helmholtz Young Investigator Program VH-NG-736.

## Additional information

### Funding

| Funder | Grant reference number | Author |
|---|---|---|
| H2020 European Research Council | SmartMic | Michael Weber Nico Scherf Jan Huisken |
| Helmholtz-Gemeinschaft | Young Investigator Program (VH-NG-736) | Alexander M Meyer Daniela Panáková |
| H2020 European Research Council | CardioNECT | Peter Kohl |

The funders had no role in study design, data collection and interpretation, or the decision to submit the work for publication.

### Author contributions

Michael Weber, Nico Scherf, Conceptualization, Data curation, Software, Formal analysis, Validation, Investigation, Visualization, Methodology, Writing—original draft, Writing—review and editing; Alexander M Meyer, Resources, Methodology; Daniela Panáková, Resources, Methodology, Writing—review and editing; Peter Kohl, Conceptualization, Investigation, Methodology, Writing—review and editing; Jan Huisken, Conceptualization, Resources, Supervision, Funding acquisition, Methodology, Writing—original draft, Project administration, Writing—review and editing

### Author ORCIDs

Michael Weber  http://orcid.org/0000-0001-8007-9975
Daniela Panáková  http://orcid.org/0000-0002-8739-6225
Jan Huisken  http://orcid.org/0000-0001-7250-3756

### Ethics

Animal experimentation: Zebrafish (Danio rerio) were kept at 28.5 °C and handled according to established protocols and in accordance with EU directive 2011/63/EU as well as the German Animal Welfare Act.

### Decision letter and Author response

Decision letter https://doi.org/10.7554/eLife.28307.024
Author response https://doi.org/10.7554/eLife.28307.025

## Additional files

### Supplementary files

• Transparent reporting form
DOI: https://doi.org/10.7554/eLife.28307.022

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
