## [Decision Letter]

Thank you for submitting your article "Cell-accurate optical mapping across the entire developing heart" for consideration by *eLife*. Your article has been reviewed by 1 peer reviewers, and the evaluation has been overseen by a Reviewing Editor and Didier Stainier as the Senior Editor. The following individual involved in review of your submission has agreed to reveal his identity: Hee Cheol Cho (Reviewer #1).

The reviewers have discussed the reviews with one another and the Reviewing Editor has drafted this decision to help you prepare a revised submission.

Summary:

Understanding the interplay among individual cells during organogenesis can provide fundamental insights into tissue-level morphogenesis during development. The quest for this essential knowledge has been elusive partly due to limitations with imaging instrumentation. In this manuscript, Huisken and colleagues adapt their pioneering technology of high-speed, high-resolution light sheet microscopy to study how conductivity evolves and matures during zebrafish heart development. Specifically, they built a high-speed, dual-color light sheet microscope to image the heart at subcellular resolution, based on a published design called post-acquisition synchronization. They then developed a set of computational tools to analyze calcium activity at cellular resolution. These include a curved cylindrical coordinate system to describe cell position along the midline, a graph-based method to calculate the conduction speed in terms of the number of cells per unit time, and a 2D projection method to display cellular properties across the 3D heart. Finally, they applied these methods to measure the emergence of region-specific activation and conduction between 36 hpf and 44 hpf that accompanies the morphological emergence of three regions of the heart.

The biological insights revealed are in line with what is known in the developing heart, including the faster Ca^2+^ rise time in atrial vs. ventricular cardiomyocytes and slower conduction velocities in proximal atrial myocytes and in AVC. These data validate that the temporal and spatial sampling rates were adequate so that the inherent lag between the beginning of biological phenomena and image capture would be irrelevant. Comparison of physical conduction velocity, afforded by this technology, and the routine biological conduction velocity revealed that the larger dimensions of atrial cardiomyocytes gave rise to faster physical conduction velocity. Altogether, this study presented a set of impressively careful technical characterization that will benefit the field. In particular, this manuscript makes a very important contribution by reconstructing the geometry of the heart, and then transforming it to a convenient "map" of the cell positions, permitting the neighbor relationships to be more easily seen and analyzed. Moreover, the manuscript is well written with beautiful figures that are well organized, intuitively informative and aesthetically appealing. However, the manuscript has several limitations in its current form (as described below) that should be addressed in a revised version.

Essential revisions:

1) The authors must be clear in their title, Abstract, and elsewhere in the text that these are hearts without excitation-contraction coupling. This is an understandable decision in the experimental design, but this does mean that the normal mechanics of the contractile tissues and the normal forces from the blood flow through the system are absent. It is probably beyond the scope of this paper to perform the study with function intact, but the authors can only argue this if they are clear in the title, Abstract, and text.

2) The authors discuss proudly the performance of the microscope used, and they show lovely data. However, beyond knowing what it is better than, the paper does not really teach the reader what the microscope is. More detail is desirable here.

3) The key steps in re-assembly of the imaging data into the volumetric rendering are underspecified, and more detail is needed.

4) The authors quantify that, strikingly, less than 10 pacemaker cells serve as the origin of electrical activation at 52 hpf. The location of these cells is the sinus venosus at the heart's inflow in a ring-like formation. The authors cite an earlier work which indicate that the myocardial cells in the outer curvature are oriented perpendicular to the inflow-outflow direction. Do the authors suggest that the pacemaker cells exhibit anisotropy at the level of individual cell morphology, linking them in a ring-like manner, and this is paralleled by neighboring atrial myocytes oriented orthogonal to the inflow-outflow line? If so, it may be easier to see this by illustrating the estimated cell shapes of the pacemaker cells as scaled ellipsoids.

5) In terms of the biology presented here, there seem to be some missed opportunities. Could the authors address how the magic transition occurs between 36 hpf and 44 hpf (e.g. gradual and smooth, or sudden appearance with chaotic transition)? Could they address whether heterogeneity exists at the cellular level (aside from the pacemaker) and, if so, whether cellular heterogeneity matters in the transition?

---

## [Author Response]

Essential revisions:1) The authors must be clear in their title, Abstract, and elsewhere in the text that these are hearts without excitation-contraction coupling. This is an understandable decision in the experimental design, but this does mean that the normal mechanics of the contractile tissues and the normal forces from the blood flow through the system are absent. It is probably beyond the scope of this paper to perform the study with function intact, but the authors can only argue this if they are clear in the title, Abstract, and text.

We agree and point this out now more prominently in the Abstract and the Introduction.

2) The authors discuss proudly the performance of the microscope used, and they show lovely data. However, beyond knowing what it is better than, the paper does not really teach the reader what the microscope is. More detail is desirable here.

We added a statement to the Results section explaining the need for a fine balance of spatial and temporal resolution and field of view – the microscope’s unique feature: “By fine-tuning the magnification and restricting camera readout to the center area of the chip, we balanced the field of view and the spatial and temporal sampling to record cardiac activation in the entire heart with cellular precision (Materials and methods)”. Furthermore, we added technical details to the “Material and methods” section under “Light sheet microscopy”.

3) The key steps in re-assembly of the imaging data into the volumetric rendering are underspecified, and more detail is needed.

We now address this point by a paragraph about the post-acquisition synchronization of movie stacks in the “Materials and methods” section, plus information on the synchronization process in the second paragraph of the “Results” part, and an additional paragraph about the visualization of the synchronized movies in the “Materials and methods” section.

4) The authors quantify that, strikingly, less than 10 pacemaker cells serve as the origin of electrical activation at 52 hpf. The location of these cells is the sinus venosus at the heart's inflow in a ring-like formation. The authors cite an earlier work which indicate that the myocardial cells in the outer curvature are oriented perpendicular to the inflow-outflow direction. Do the authors suggest that the pacemaker cells exhibit anisotropy at the level of individual cell morphology, linking them in a ring-like manner, and this is paralleled by neighboring atrial myocytes oriented orthogonal to the inflow-outflow line? If so, it may be easier to see this by illustrating the estimated cell shapes of the pacemaker cells as scaled ellipsoids.

To clarify this point, we add new imaging data that highlight cell orientations in the developing heart at 52 hpf, using a novel fluorescent reporter expressed in myocardial cell membranes (*myl7*:lck-eGFP) in combination with the cardiomyocyte-specific fluorescent nuclei reporter (*myl7*:H2A-mCherry): see Figure 1—figure supplement 8. Our new data shows clearly that cells at the atrial inflow site are indeed elongated and arranged in a ring-like fashion, perpendicular to the cardiac axis. It also illustrates that ventricular cells – which multiply rapidly at this stage – are much smaller, explaining the shallower interrelation between biophysical (distance based) and biological (cell number based) conduction velocities (Figure 1—figure supplement 6). We describe the sample preparation and imaging modalities used to record the new data in the two new sections “Sample preparation for confocal and two-photon microscopy” and “Confocal and two-photon microscopy” in the “Materials and methods” part. We have also added two authors crucially involved in gathering this information: Alexander M. Meyer and Daniela Panáková who developed the transgenic *myl7*:lck-eGFP zebrafish line.

5) In terms of the biology presented here, there seem to be some missed opportunities. Could the authors address how the magic transition occurs between 36 hpf and 44 hpf (e.g. gradual and smooth, or sudden appearance with chaotic transition)? Could they address whether heterogeneity exists at the cellular level (aside from the pacemaker) and, if so, whether cellular heterogeneity matters in the transition?

Based on our image data at those two discrete time points, it is difficult to be certain about the dynamics of the transition. Judging by the calcium transient data in Figure 2 and the changes in regional conduction velocities (Figure 2—figure supplement 2) we would argue that the transition is gradual. We made this clearer in the revised manuscript by inserting relevant descriptions, e.g. “The gradual formation of atrial (1) and ventricular (2) populations with increasingly different calcium transients is highlighted by the dashed lines at 52 hpf.” and “A progressive crowding of calcium transient activation dynamics indicated maturation of cells, with two groups differentiating from the early more homogeneous pattern: working cardiomyocytes in atrium and ventricle…”.

A key biological aspect, in our view, is the fact that it is not the pacemaker and AVC ‘nodal equivalents’ that progressively specialize during development (as is often implied in adult cardiac cell electrophysiology research), but the atrial and ventricular working myocardium. This is stated as follows: “Myocardial cells in the two chambers remodel and specialize into functional tissue of working atrial and ventricular cardiomyocytes, while cells in the pacemaker and AVC regions continue to resemble the earlier phenotype from the tubular stage.”